# Profiling Walnut Fungal Pathobiome Associated with Walnut Dieback Using Community-Targeted DNA Metabarcoding

**DOI:** 10.3390/plants12122383

**Published:** 2023-06-20

**Authors:** Marie Belair, Flora Pensec, Jean-Luc Jany, Gaétan Le Floch, Adeline Picot

**Affiliations:** Laboratoire Universitaire de Biodiversité et Ecologie Microbienne, INRAE, University Brest, F-29280 Plouzané, France

**Keywords:** metabarcoding, internal transcribed spacer (ITS), mock communities, environmental DNA (eDNA), walnut dieback, fungal pathogens

## Abstract

Walnut dieback can be caused by several fungal pathogenic species, which are associated with symptoms ranging from branch dieback to fruit necrosis and blight, challenging the one pathogen–one disease concept. Therefore, an accurate and extensive description of the walnut fungal pathobiome is crucial. To this end, DNA metabarcoding represents a powerful approach provided that bioinformatic pipelines are evaluated to avoid misinterpretation. In this context, this study aimed to determine (i) the performance of five primer pairs targeting the ITS region in amplifying genera of interest and estimating their relative abundance based on mock communities and (ii) the degree of taxonomic resolution using phylogenetic trees. Furthermore, our pipelines were also applied to DNA sequences from symptomatic walnut husks and twigs. Overall, our results showed that the ITS2 region was a better barcode than ITS1 and ITS, resulting in significantly higher sensitivity and/or similarity of composition values. The ITS3/ITS4_KYO1 primer set allowed to cover a wider range of fungal diversity, compared to the other primer sets also targeting the ITS2 region, namely, GTAA and GTAAm. Adding an extraction step to the ITS2 sequence influenced both positively and negatively the taxonomic resolution at the genus and species level, depending on the primer pair considered. Taken together, these results suggested that Kyo set without ITS2 extraction was the best pipeline to assess the broadest fungal diversity, with a more accurate taxonomic assignment, in walnut organs with dieback symptoms.

## 1. Introduction

Walnut (*Juglans regia* L.) cultivation is one of the most important cultivations of nut crops worldwide, reaching more than 3.5 billion tons and 1.1 million hectares in 2021 [1]. China is the leading walnut producer, followed by the United States and Iran, while Europe ranks 5th [1]. Walnut orchards are usually affected by various pathogens, including *Xanthomonas campestris* pv. *juglandis*, which causes walnut blight, *Ophiognomonia leptostyla* and species from the *Colletotrichum acutatum* species complex, both responsible for walnut anthracnoses, as well as *Geosmithia morbida*, the causal agent of the “Thousand canker disease” [2,3,4,5]. Walnut dieback has also been frequently reported in California [6] as well as in Mediterranean-climate countries such as Spain [6,7], Italy [8], Turkey [9], and Czech Republic [10] over the past decade and more recently in France [11]. This fungal disease is characterized by symptoms such as fruit necrosis and blight, twig defoliation and dieback as well as branch canker up to host death [6,7,8,9,10,11]. The *Botryosphaeriaceae* family, mainly represented by the *Botryosphaeria*, *Diplodia* and *Neofusicoccum* genera, and *Diaporthe* spp. (teleomorph of *Phomopsis* spp.) were described as the causal agents in these countries; however, other pathogenic fungi such as *Colletotrichum* spp. and *Fusarium* spp. could be present in smaller proportions and participate in walnut decay and twig dieback [6,7,11,12,13]. Walnut dieback is thus likely to be caused by complexes of species working synergistically and/or antagonistically to create, foster or mitigate disease onset and development as it has been demonstrated for dieback diseases affecting other tree crops such as vines [14]. As such, this disease is an excellent illustration of the pathobiome concept that can be defined as “the set of host-associated organisms associated with reduced (potentially reduced) health status, as a result of interactions between members of that set and the host” [15]. It is therefore important to investigate and thoroughly describe the fungal pathogens associated with these new symptoms in walnut orchards without neglecting the contribution of other members of the microbiota to disease development. To do so, environmental DNA (eDNA) metabarcoding represents a powerful culture-independent technique to decipher the mycobiota of affected walnut organs as well as their associations and thus gain access to the pathobiome functioning [16,17,18,19,20].

Metabarcoding has been extensively used to assess microbial diversity of various agroecosystems [21,22,23,24]. Recently, this method has been used to evaluate fungal diversity associated with leaves [25] and buds [5] of diseased walnut trees. Metabarcoding success and accuracy depend on the selection of the targeted region, the design of primer pairs, and the choice of adapted bioinformatic analysis pipelines [26,27,28,29]. It is then important to define an adapted and tailored analysis method for each ecosystem, considering the biases that may arise at each step of eDNA metabarcoding [26] from polymerase chain reaction (PCR) parameters and amplification [30,31,32,33] to taxonomic assignment of final reads during bioinformatic analysis [34]. The internal transcribed spacer (ITS) region is the universal barcode used to study fungal communities [35], and many studies have designed universal primers for the amplification of this region for a large number of fungal species [36,37,38,39,40,41,42,43,44,45].

In order to validate an analysis method, a mock community control is highly recommended [26,46]. It is composed of DNA solutions of known organisms, likely to be encountered in the targeted ecosystem, and mixed in known proportions. This positive control enables to evaluate and estimate numerous experimental, sequencing and bioinformatic analysis biases during sample preparation and DNA extraction, PCR sequence amplification (e.g., over- or under-representation of taxa and chimera formation), and taxonomic assignment [28,46,47,48,49].

To the best of our knowledge, our study is the first to validate metabarcoding protocols and assess fungal pathogen diversity associated with walnut diseases based on mock communities. The aim of this study was to define a metabarcoding methodology enabling the characterization of walnut fungal pathobiome associated with walnut dieback disease with an accurate taxonomic resolution up to the genus or species level. The validated method was subsequently applied on symptomatic husk and twig samples from walnut orchards.

## 2. Results

### 2.1. Analysis of Mock Communities and Identification of the Best Primer Sets

To evaluate the performance of each combination of primer pairs and pipelines in characterizing fungal mycobiota associated with symptomatic walnut organs, five mock communities, comprising between 5 to 18 fungal pathogenic and endophytic/saprophytic walnut-associated DNA species, were sequenced using Illumina MiSeq metabarcoding.

After read filtering, a total number of 3,509,971 sequence reads were obtained from 15 mock community samples (5 mocks × 3 replicates) and all primer pairs, with a mean number of sequences per sample ranging from 32,318 ± 5232 (GTAAm) to 78,792 ± 18,049 (Kyo) and a retention percentage of at least 41.6% after read trimming and filtering (Appendix A). Rarefaction curves plotting the sequencing depth against the number of ASVs showed that the plateau phase was reached for every community and every primer set (Appendix A).

The distribution of genera showed a high repeatability between replicates irrespective of the mock and primer set (Figure 1). In addition, walnut DNA did not affect the quality of metabarcoding sequencing as shown by the similar distribution between Mock3 and Mock5 (Figure 1). Furthermore, similarity of composition values were not significantly different between these two mock communities (*p* = 0.2997, Kruskal-Wallis test).

All primer sets demonstrated the ability to detect *Botryosphaeriaceae* and *Diaporthe* species when these species were not mixed with others (Mock1 and Mock2) as shown by the high sensitivity values (Table 1 and Table 2). However, they did not detect the complete diversity of Mock3, Mock4 and Mock5. The ITS1 primer set detected the lowest number of genera (up to 8 species out of 18 in at least one replicate of Mock4), and GTAA failed to amplify *Colletotrichum godetiae*, and its ability to detect *C. fioriniae* was relatively low (between 60 and 116 reads depending on replicate; Table 1). Moreover, considering the most diverse mock community (Mock4), no significant correlations were found between obtained and expected relative abundances for ITS (R = 0.012 and *p* = 0.95) and ITS1 sets (R = −0.17 and *p* = 0.35) mainly because of an overrepresentation of *Fusarium*, *Epicoccum* and *Gibellulopsis* genera and an underrepresentation of *Botryosphaeriaceae* and *Diaporthe* species (Figure 1 and Figure 2).

Compared to ITS and ITS1 primer pairs, the three other sets targeting the ITS2 barcode showed overall better performance in terms of sensitivity and/or similarity of composition; however, Kyo was associated with the lowest precision values along with ITS. This discrepancy is mainly caused by metabarcoding sequencing with these primer pairs that generated supernumerary ASVs, thus overestimating the number of ASVs obtained compared to the expected number of ASVs and leading to low precision values at the ASV scale. First, GTAAm degenerated primer pair permitted the amplification and detection of *Colletotrichum* in addition to other genera of interest when compared with GTAA (Figure 1C,D). GTAAm was also associated with significantly higher sensitivity values than ITS1 when considering Mock4 (Table 2). Overall, both GTAAm and Kyo showed the highest positive and significant correlation coefficients between observed and expected relative abundances for Mock4 (R = 0.51; *p* = 2.5 × 10^−3^ and R = 0.6; *p* = 2.4 × 10^−4^ respectively; Figure 2). Nonetheless, both primer pairs underestimated or overestimated the relative abundance of a few studied genera. In particular, relative abundance of *Lasiodiplodia* was underestimated by both GTAAm and Kyo (with a ratio of expected to recovered relative abundances of 2.254 ± 0.079 and 2.603 ± 0.009, respectively) as well as the *Botryosphaeria* (1.838 ± 0.047 and 2.080 ± 0.058, respectively) and *Diaporthe* relative abundances (1.908 ± 0.041 and 1.711 ± 0.050, respectively). *Alternaria* was underestimated by GTAAm (3.183 ± 0.146), and *Fusarium* was overestimated by both GTAAm (0.383 ± 0.003) and Kyo (0.468 ± 0.005; Figure 1 and Figure 2).

Given that GTAA failed to amplify *Colletotrichum* species, which are important pathogens of interest in walnut, while ITS and ITS1 provided poor similarity of composition values, only GTAAm and Kyo were utilized for the following steps, i.e., optimization of bioinformatic analyses and analysis of environmental samples.

### 2.2. Evaluation of the Taxonomic-Level Resolution

The genus- and species-level resolution of regions targeted by GTAAm and Kyo was assessed using phylogenetic trees based on ITS2 sequences of wood and fruit-associated pathogens.

First, phylogenetic trees with Bayesian posterior probabilities (BPP) were constructed based on our local amplicon databases, subjected or not to an extraction of the ITS2 region, in order to evaluate the impact of an extraction step with ITSx software on the quality of taxonomic resolution. Taxonomy of the fungal species included in mock communities was manually inspected and validated up to the species or genus level if nodes before clade containing targeted species or genus were supported by relatively strong BPP values (>0.70, considering that the analysis is based only on 1 locus; Table 3). At the species level, the highest assignment rates with both primer pairs were obtained without ITS2 extraction for Kyo (9 out of 18 species included in mock communities) and with ITS2 extraction for GTAAm (5 out of 18; Table 3). The main issues resulted from a few *Dothiorella* species and *N. mediterraneum* that were clustered together (in all conditions), *N. parvum* that was often clustered with the previous ones (except for Kyo without ITSx extraction condition) and *B. dothidea* that could not be differentiated from *Neoscytalidium dimidiatum* in most conditions (except for GTAAm with ITSx extraction condition; Table 3). These species were therefore given new assignments considering their possible multiple assignments (e.g., *N. parvum* was reassigned as *Neofusicoccum/Dothiorella* and *B. dothidea* as *Botryosphaeria/Neoscytalidium* for the relevant conditions). Moreover, *L. theobromae* sequences were always mixed with other *Lasiodiplodia* species, while *F. juglandicola* sequences were mixed with *F. avenaceum* ones.

Second, additional phylogenetic trees were built by adding ASVs sequences (from metabarcoding of mock samples) to our local database to determine their taxonomic placement within the tree and whether it was in accordance with their assignment after a blast against our ITS local database. We found that ASVs were generally grouped with the corresponding reference sequences at the genus or species level, regardless of the combination of primer pairs and pipelines (Figure 3).

The manually-corrected taxonomy of ASVs, as described above, was also used to compare the performance criteria of the two selected primer pairs depending on the addition of an extraction step of the ITS2 sequence. This step did not affect the mean number of reads after filtering for all mock communities for the two primer pairs, suggesting that no aspecific amplifications occurred (Appendix A). Thus, the average number of ASVs using mock community remained the same as well as the similarity of composition at the genus and species level, which was significantly higher with Kyo for Mock3 to Mock5 (Table 4).

In contrast, sensitivity at the genus level was significantly higher with GTAAm than with Kyo when a step of ITS2 extraction was added, while the opposite was found without the ITS2 extraction step (Table 4). In any case, at the species level, the highest sensitivity rate was obtained with Kyo without ITS2 extraction, further confirming that these conditions were the best to correctly assign the fungi of interest at this taxonomic rank (Table 4). Nonetheless, the precision criterion at the genus level remained significantly higher with GTAAm with ITS2 extraction than with Kyo (with (*p* = 3.2 × 10^−4^) and without (*p* = 0.008) ITS2 extraction).

### 2.3. Application on Environmental Samples

Based on these results, the two combinations GTAAm with ITS2 extraction and Kyo without ITS2 extraction were then applied on symptomatic environmental samples. The samples were collected from three French walnut orchards (P7, P10 and P12) and matched with symptomatic twigs (T) or husks (H). Taxonomic assignment of ASVs corresponding to phytopathogenic fungi of interest (i.e., fungal taxa introduced in mock communities) was manually inspected and modified as described above. Taxonomic assignment of the other fungal taxa was manually corrected at the genus level, or family level when necessary, based on the first 150 hits obtained with BLASTn.

After read filtering, ITS2 metabarcoding sequencing of the three replicates of six walnut samples yielded a total of 1,151,054 read sequences with Kyo (63,947 ± 33,362 mean reads per sample clustered into 108 ± 23 ASVs), while 693,700 read sequences (38,539 ± 10,832 mean reads per sample clustered into 49 ± 17 ASVs) were obtained with GTAAm. Replicate P7-H-2020-3 was associated with only 173 raw sequence reads when amplified with GTAAm and was discarded from further analysis. Combining results from husks and twigs, 25 and 24 genera were obtained for GTAAm and Kyo, respectively, of which 18 were in common. Each combination of primer pair and pipeline enabled the detection of the pathogenic genera in walnut trees, i.e., *Botryosphaeria/Neoscytalidium*, *Neofusicoccum/Dothiorella*, *Colletotrichum*, *Diaporthe*, *Fusarium*, *Juglanconis* and *Phaeoacremonium* (Figure 3 and Figure 4). Unlike Kyo, GTAAm did not permit the detection of *Epicoccum* genus as well as *Basidiomycota* taxa (*Bulleribasidiaceae* family mainly represented by *Vishniacozyma* genus and to a lesser extent unidentified genera of *Sirobasidiaceae* family and *Tremellales* order). Moreover, Kyo showed a better taxonomic resolution for certain genera within the *Pleosporales* order than GTAAm. The genera *Angustimassarina* and *Biatriospora* were identified with Kyo but were reassigned at the order level (unidentified_*Pleosporales*) with GTAAm (Figure 4).

For both primer sets, alpha-diversity indices, including Shannon and InvSimpson indices calculated at the genus level, were not significantly different irrespective of the index and the type of organ studied. Nonetheless, the number of detected genera (observed index) associated with environmental husk samples was significantly higher with Kyo than with GTAAm, further highlighting a better capacity of Kyo to cover a wider range of diversity (*p* = 0.004; Figure 5).

## 3. Discussion

The aim of this study was to define both metabarcoding parameters and bioinformatic pipeline to profile the fungal pathobiome associated with walnut dieback symptoms in eDNA samples. In order to choose the best combination of barcode and pipeline, we assessed the ability of five primer pairs to detect, assign and accurately estimate the relative abundance of fungal strains commonly associated with walnut or walnut dieback included in mock communities using Illumina MiSeq PE 300 bp sequencing. The selection of targeted barcodes, primers and bioinformatic pipelines is crucial in light of their significant influence on the resolution of taxonomic assignment [29,50,51,52,53]. The five primer pairs, all targeting whole or a part of the ITS region designated as the fungal universal barcode [35], i.e., ITS1, ITS2 and ITS, were compared using the DADA2 bioinformatic tool. Here, targeting the ITS2 region was shown to improve similarity of composition and/or sensitivity compared to ITS and ITS1. In addition, the quality of the analysis also depends largely on bioinformatic pipelines that must be tailored according to the fungal taxa expected to be associated with the considered ecosystem. Many studies have compared various barcodes for optimal fungal community analyses, which evidenced a lack of consensus in the choice of the best barcode between ITS1, ITS2 or the whole ITS region probably because it notably depends on the fungal taxa that are being considered [20,54,55,56,57,58]. For instance, the GTAA primer pair was set aside in this study because of its inability to amplify *Colletotrichum* species, which were not part of the tested fungal species by Morales-Cruz et al. [20]. The degeneration of the forward primer of this barcode enabled to solve this issue as suggested by Tedersoo et al. (2016) to minimize primer bias [29].

Interestingly, two primer pairs targeting the ITS2 region, GTAA182fm/GTAA526r and ITS3/ITS4_KYO1, provided the most accurate identifications and estimations of relative abundances and were selected for further analyses. On the basis of phylogenetic trees, both sets showed the same overall ability in distinguishing fungi of interest at the genus level, while Kyo showed better taxonomic resolution at the species level without the use of ITSx. This manual inspection step showed that care should be taken with the ASVs assignment just after bioinformatic pre-processing as it strongly depends on the taxonomic assignment algorithm as well as on the database even when UNITE database, the largest curated database dedicated to ITS sequences, is used [53,59]. Note that the same analyses were performed with the UNITE+INSDC non-redundant fungal ITS database v8.3 (1,039,010 ITS fungal sequences) before the latest version was released. The accuracy of the taxonomic assignment was much lower than those obtained with the v9.0 database (6,441,764 ITS fungal sequences). Thus, the number of sequences in the database, the diversity it covers, and the quality of the sequences are other crucial parameters in the quality of metabarcoding sequencing bioinformatic analyses [60,61,62]. The use of the latest version of a reference database seems to be essential, especially in view of the increase of knowledge about microbial diversity and of available sequences. The manual curation step allowed for the reassignment of ASVs to a more rightful taxonomic rank, notably those initially assigned to *Botryosphaeria dothidea* and *Neofusicoccum* spp. that were found clustered with a few other genera in the phylogenetic tree and therefore reassigned accordingly (Table 3).

Although Kyo and GTAAm provided high similarity of composition values, there were a few discrepancies between expected and observed relative abundances, up to a factor of 3. This has been consistently reported in many studies, and it was hypothesized that ITS length polymorphism likely accounted for such differences [63,64]. In other words, shorter amplicons are more likely to amplify during PCR [65] resulting in overrepresented taxa and, consequently, due to the compositional nature of metabarcoding data, underrepresented taxa [66]. Additional polymorphism form that could lead to an overestimation of taxa is the copy number variation (CNV) of the targeted region. Recently, Lofgren et al. (2019) inferred using in silico analysis of fungal genomes that the CNV of ribosomal DNA containing the ITS region could vary from 14 to 1442 copies among 91 fungal taxa belonging to *Ascomycota*, *Basidiomycota*, *Chytridiomycota*, *Mucoromycota* and *Zoopagomycota* phyla [67]. Another main bias associated with PCR is that more abundant taxa are more likely to be amplified [66], which was actually not evaluated in our mock communities mixed in even proportions. Nanopore and PacBio sequencing technologies do not require a PCR step and may thus overcome these issues. In addition, sequencing longer reads allow a more accurate taxonomic assignment. However, the use of these technologies are still limited probably because of their high cost.

Furthermore, extraction of the variable ITS1 or ITS2 regions from raw sequences is recommended in many pipelines in order to discard conserved flanking sequences and thus improve species identification [51,68,69]. Based on this study, the interest of adding an extraction step depended on the primer set tested. Actually, extraction of the ITS2 region improved sensitivity with GTAAm, while the opposite was found with Kyo, and most probably was because of the loss of genetic information in sequences obtained with Kyo in the 28S region. Finally, we evaluated the capacity of the combinations of the two selected primer pairs with and without ITS2 extraction to describe well the symptomatic environmental samples. All conditions tested successfully permitted the detection and amplification of the major fungal phytopathogens and endophytes or saprophytes associated with walnut dieback and trunk diseases worldwide, namely, *Botryosphaeria*, *Colletotrichum*, *Diaporthe*, *Fusarium*, *Neofusicoccum*, *Juglanconis* and *Phaeoacremonium* genera [5,10,25,70,71,72]. Alpha-diversity indices were not significantly different between the two primer pairs except for the number of detected genera associated with walnut husk samples, which was significantly higher with Kyo. Note that only Kyo led to the amplification of *Basidiomycota* sequences, providing a broader view of fungal diversity of the pathobiome in the environmental samples [38].

## 4. Materials and Methods

### 4.1. Identification of Primer Sets Targeting Walnut Pathogens

The internal transcribed spacer (ITS) of the ribosomal DNA region was chosen as target barcode to identify fungal communities. A local database was built with full length ITS sequences of 78 species from 17 genera commonly isolated from walnut, almond, pistachio and olive trees worldwide, i.e., *Geosmithia, Botryosphaeria, Diplodia, Dothiorella*, *Lasiodiplodia*, *Neofusicoccum*, *Neoscytalidium*, *Diaporthe*, *Epicoccum*, *Colletotrichum*, *Ophiognomonia*, *Juglanconis, Fusarium*, *Gibellulopsis*, *Alternaria*, *Phaeoacremonium* and *Cytospora* (Table 5). The database was generated by downloading ITS sequences from the Genbank database [73] of National Center for Biotechnology Information [74] (accessed on 21 May 2021) for each fungal species using the query word ‘internal transcribed spacer’. Up to 10 sequences per species were retrieved as well as the species reference sequence (RefSeq) when available [75].

Three universal primer sets amplifying the whole ITS region (ITS1F/ITS4, hereafter ITS [36,37]), the ITS1 region (ITS1F/ITS2, hereafter ITS1 [36,37]) and the ITS2 region (ITS3/ITS4_KYO1, hereafter Kyo [36,38]) regions were selected. Two primer sets were added because of their ability to specifically amplify part of the pathogens of interest (GTAA182f/526r and GTAA182fm/526r, hereafter GTAA and GTAAm, respectively) (Table 6 and Figure 6). GTAA was designed by Morales-Cruz et al. [20] to amplify grapevine-associated pathogens using metabarcoding sequencing, and some of which are common to those isolated from nut crops (i.e., *Botryosphaeria*, *Diplodia*, *Dothiorella*, *Lasiodiplodia*, *Neofusicoccum*, *Diaporthe* and *Phaeoacremonium*). In silico amplification of the local ITS database with the GTAA182f/526r primer pair was performed using Geneious Prime 2021.1.1 (settings: pairs only anywhere on the sequences and two mismatches allowed except within 15 bp of 3′ end; https://www.geneious.com (accessed on 21 May 2021). Since primers GTAA182f/526r failed to amplify the targeted region in *Colletotrichum* spp. because of a mismatch in the forward primer sequence, it was modified by replacing the C base in the 9th position by Y and renamed GTAA182fm.

### 4.2. Fungal Mock Communities

Taxonomic identity of isolates used to construct mock communities were preliminarily checked by amplifying the region targeted by the ITS4/ITS5 primer pair, following PCR conditions described in White et al. [36]. Sanger sequencing of amplicons was performed by Eurofins Genomics platform (Cologne, Germany). Contig assembly and taxonomic assignment were performed with Geneious Prime 2021.1.1.

Five mock communities (Mock1–Mock5) were generated by combining DNA solutions in equal concentrations from pure cultures of walnut fungal pathogens isolated from symptomatic walnut husks and twigs collected in French walnut orchards. In addition, cultures from the Westerdijk Fungal Biodiversity Institute (CBS collection, Utrecht, Netherlands) and the University of Western Brittany Culture Collection (UBOCC, Plouzané, France) were also included to cover the range of pathogenic fungal species associated with walnut decay and necrotic twigs reported in several countries (Table 7). DNA extracted from asymptomatic walnut husks was added in Mock5 to evaluate the potential matrix effect on species amplification during PCR given the high levels of potential PCR inhibitors, including polyphenols, in husks [140,141] (Table 7). Besides known walnut pathogens, mock communities were also composed of DNA from non-phytopathogenic fungi frequently isolated in symptomatic organs during preliminary tests, i.e., *Gibellulopsis nigrescens* and *Epicoccum nigrum*. Pure fungal cultures and mock communities DNA concentrations were quantified with a Quantus™ Fluorometer and a QuantiFluor^®^ ONE ds DNA System (Promega Corporation, Madison, WI, USA). Three replicates were prepared for each mock community.

### 4.3. Sampling and Total DNA Extraction

Symptomatic walnut husks (five per orchard) and twigs (12 per orchard) from different trees were collected from three French orchards (P7, P10 and P12) located in the Southwest of France in September 2020 and May 2021, respectively. Symptoms on walnut husks were characterized by blight and necrosis, while symptomatic twigs showed necrosis and dieback symptoms. Organs were surface-sterilized as follows: 1 min in a 2% active chlorine bleach solution and 1 min in two sterile distilled water baths before drying in sterile filters. Bark of twigs was removed, and the border between healthy and necrotic tissues was cut from twigs and walnut husks using scalpel. Then, samples were pooled by orchard and by sample type (husk or twig) before being lyophilized for 48 h (Freeze-dryer Alpha 1-4 LDplus©, Martin Christ Gefriertrocknungsanlagen GmbH, Osterode am Harz, Germany) and ground in steel jars using Retsch MM400 (Retsch GmbH, Haan, Germany) until fine powder was obtained. Environmental DNA extractions were performed in triplicate using FastDNA™ SpinKit (MP Biomedicals, Fisher Scientific, Waltham, MA, USA) following the manufacturer’s instructions and quantified with a NanoDrop 1000 Spectrophotometer (Thermo Fisher Scientific, Waltham, MA, USA). To monitor potential contamination, extraction blanks were prepared alongside the samples.

### 4.4. Illumina MiSeq Sequencing and Sequence Analyses

Prior to sequencing, evaluation of success of PCR amplification was performed on DNA from mock communities and from walnut husks and twigs using the GTAA182f/526r, GTAA182fm/526r and ITS3/ITS4_KYO1 primer pairs. PCR stages were based on the protocol described in Morales-Cruz et al. [20] for the two initial primer pairs and in Toju et al. [38] for the latter, with a volume of 0.2 µL of DNA solution at a concentration of 20 ng/µL. All PCR were performed with a Doppio thermal cycler (VWR™) and GoTaq^®^ G2 Flexi DNA Polymerase kit (Promega Corporation) but without BSA. The quality of the PCR products was checked using electrophoresis on 1% agarose gels with Tris-acetate-EDTA (TAE) buffer 1X (Promega Corporation) and Midori Green Advance^®^ stain (Nippon Genetics Co. Europe GmbH, Düren, Germany). No PCR products were detected for extraction blanks.

Amplicon libraries and Illumina MiSeq PE 300 bp sequencing were performed in the same run at the McGill University and Génome Québec Innovation Center (Montréal, Canada) with same adapter FLD_ill (forward sequence: ACACTCTTTCCCTACACGACGCTCTTCCGATCT; reverse sequence: GTGACTGGAGTTCAGACGTGTGCTCTTCCGATCT), and the PCR conditions are listed in Appendix A.

Sequence analysis workflow is depicted in Figure 7. First, we used FIGARO [142] to determine optimal trimming parameters of forward and reverse reads with the best percentage of read retention [143] (estimated amplicon length set to 450 bp and minimum overlap length of 20 bases; Appendix A) followed by the DADA2 R package [144] for read truncation with no ambiguous bases allowed, read denoising and filtering. Quality profiles were also inspected with DADA2 R package and reads were trimmed according to Figaro parameters allowing the inclusion of bases with a minimum Qscore of 30 (corresponding to a probability of an incorrect base call 1 in 1000 times). In the case of ITS, forward and reverse reads were non-overlapping. Therefore, optimal trimming parameters of the forward and reverse reads were assessed using FIGARO with an estimated simulated amplicon length of 450 bp without minimum overlap length (Appendix A). Sequences were then concatenated by adding N bases between forward and reverse reads.

For all primer pairs, amplicon sequence variants (ASVs) were independently inferred from the forward and reverse reads of each sample using the run-specific error rates, and read pairs were merged with an overlap setting of 12 bases minimum, except for ITS reads which were concatenated using the “*justConcatenate = TRUE*” argument. Then, taxonomic assignment was performed with BLASTn command line [145] against the UNITE+INSDC non-redundant fungal ITS v9.0 database [59], and the first hit was extracted [35], and then, taxonomic assignment was manually checked and corrected by checking the first 150 hits for each ASV. In addition, a BLASTn against the ITS local database was performed for ASVs associated with species included in mock communities (details in Section 2.1). To avoid overrepresentation of rare ASVs, only those represented in at least two samples and with sequence count greater than 10 were included in community analysis [27,31,146] using the *filter_otus_from_otu_table.py* QIIME [147] script.

In order to improve taxonomic resolution, the interest of extracting ITS2 region was evaluated for both GTAAm and Kyo by adding an ITSx [68] extraction step after read merging (Figure 7).

Blank PCR samples provided by the metabarcoding sequencing platform were added to the analyses, and no sequences were detected after the pre-processing step.

Processing analyses were performed using Phyloseq R package [148]. For the alpha-diversity indexes, rarefaction was applied to normalize all datasets at the same read counts based on the smallest samples across all datasets (sample size of 18,201 reads).

### 4.5. Comparative Evaluation of Primer Pairs

Four performance criteria, namely, sensitivity, precision and similarity of composition and number of ASVs were used to evaluate the performance of the five primer pairs [28]. Sensitivity was defined as the ability of a primer pair to assign well an ASV at genus level and was calculated as TP/(TP + FN), where TP corresponds to the number of true positive and FN the number of false negative ASVs (i.e., either not detected or detected but wrongly assigned at the genus level). Precision was defined as the quality of detection of fungal species and was calculated as TP/(TP + FP), where FP indicates the number of false positive ASVs (i.e., either supernumerary ASVs or ASVs assigned to unexpected genera). Similarity of composition between the recovered community and the expected community was defined as 1-BC, where BC is the Bray–Curtis similarity index and calculated as in Pauvert et al. [28]. To calculate this last criterion, taxonomic assignment obtained against the local ITS database was used. Last, the mean number of ASVs obtained for each combination of primer pair and pipeline was also determined.

### 4.6. Evaluation of the Taxonomic-Level Resolution

A database targeting the ITS2 region (and including 5.8S ribosomal RNA and large subunit (LSU) RNA) was constructed for GTAAm and Kyo. This local database was generated by downloading culture collection sequences from the Genbank database [73] of National Center for Biotechnology Information [74] (accessed on 17 February 2023) for each pathogenic fungal species commonly isolated from nut crops and olives (Table 5) using the query word ‘5.8S’ with a sequence length from 200 to 5000 bp. The final database was composed of 3596 sequences for a total of 77 species. The total number of sequences per species was listed in Appendix A.

To assess the taxonomic resolution of the ITS2 region targeted by GTAAm and Kyo primer pairs, phylogenetic trees were then built using this local database. First, amplicons were extracted for each primer pair following the same settings in Geneious Prime 2021.1.1, as described previously. If no amplicons were obtained in silico with these parameters, mismatches were allowed anywhere on the primer sequences (settings: pairs only anywhere on the sequences and two mismatches allowed). For each species, only unique sequences were retained by removing duplicate sequences. These selected sequences, in which taxonomic identification was manually checked using Nucleotide BLAST (https://blast.ncbi.nlm.nih.gov/Blast.cgi (accessed on 10 November 2021)), were used to build phylogenetic trees. The Kyo amplicon database contained a smaller number of species because of the difficulty in retrieving sequences long enough for reverse primer attachment. Moreover, no long enough ITS sequences of *Dothiorella omnivora* from culture collections were available to allow amplification by the two primer pairs (Appendix A).

Then, final databases were aligned using MAFFT version 7 online service [149] with the Auto algorithm for alignment strategy. The resulting alignments were edited using Gblocks 0.91b [150,151] with all the options to allow less stringent selection. To evaluate the effect of ITS region extraction on primer pair resolution, the ITS2 region was extracted from amplicons of GTAAm and Kyo using ITSx [68], and phylogenetic trees were built based on extracted or non-extracted local databases. Phylogenetic trees were then constructed based on Bayesian inference with the Bayesian Evolutionary Analysis Sampling Trees 2 (BEAST2) version 2.6.1 package [152]. The best models were estimated for each amplicon database using the bModelTest add-on in Bayesian Evolutionary Analysis Utility (BEAUti2) simultaneously with the Bayesian inference analysis [153]. The analysis was performed using the Markov Chain Monte Carlo (MCMC) method by performing seven independent repetitions of 100,000,000 generations, and each sampling a tree at every 1000 generations.

Convergence of the independent Bayesian inference analyses was checked using Tracer v1.7.1 software [154] and the obtained files were combined with the LogCombiner program (frequency of resampling of 100,000 and burn-in fixed at 10%). Posterior probabilities of consensus trees were determined using the Treeannotator program and visualized using FigTree v1.4.4 [155].

Finally, taxonomic placement of ASVs from mock communities and environmental samples was checked using phylogenetic reconstructions performed following the same steps after adding the ASVs sequences to the local databases.

### 4.7. Statistical Analyses

Kruskal–Wallis test followed by Dunn’s test were used to compare performance criteria obtained with the different primer sets and to evaluate the interest of ITSx extraction at the genus and the species level for each primer pair. Wilcoxon test without *p*-value correction was used to compare similarity of composition values between the two ITS2 region targeting primer pairs (GTAAm and Kyo) for each mock community. Expected and recovered relative abundances were calculated at the genus level by dividing the expected and recovered abundances by the total abundance obtained for each replicate. Simple linear regression between recovered and expected relative abundances were determined and Pearson correlation coefficient (R) and *p*-value were calculated. Alpha-diversity indices, after rarefaction based on the lowest number of reads (see Section 4.4) were compared using Wilcoxon test without *p*-value correction.

## 5. Conclusions

In conclusion, these results indicate that targeted barcode and primer pairs greatly differ in their ability and accuracy to assess the relative abundance of fungi in a metabarcoding-sequenced community. This study provided recommendations on bioinformatic analyses and primer performance for metabarcoding sequencing of environmental samples of walnut and other nut and olive trees. Depending on the targeted species, the desired taxonomic resolution (at the genus or species level) and the scale within fungal communities (the pathobiome or the phytomicrobiome), the two GTAAm and Kyo primer sets associated with the DADA2 tool and the ITSx software are good candidates. Further work could involve a metabarcoding sequencing of other environmental samples from walnut trees to better characterize fungal communities and particularly pathobiome among different locations and assess the interactions between these fungi and the associated phytomicrobiome. Although our study was tailored for walnut samples, it could certainly be applied to almond, pistachio and olive trees and also any plant samples contaminated by these pathogens provided that local database is enriched with the plant-associated fungi of interest that were not included here.

## Figures and Tables

**Figure 1 plants-12-02383-f001:**
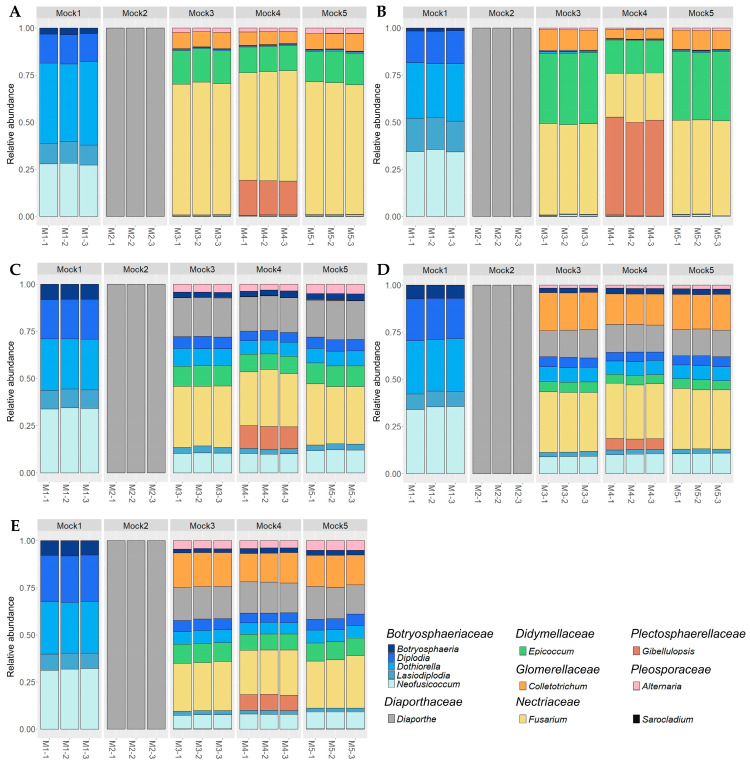
Histograms showing the relative abundance at the genus level for each replicate (−1, −2, −3) of mock communities (M1, M2, M3, M4, M5) according to metabarcoding sequencing based on the five primer sets: ITS (**A**), ITS1 (**B**), GTAA (**C**), GTAAm (**D**) and Kyo (**E**). Genera are ordered by family taxonomic rank in the legend. The *Sarocladium* genus is shown separately because it is not a component taxon of the mock communities and is identified as a contaminant genus.

**Figure 2 plants-12-02383-f002:**
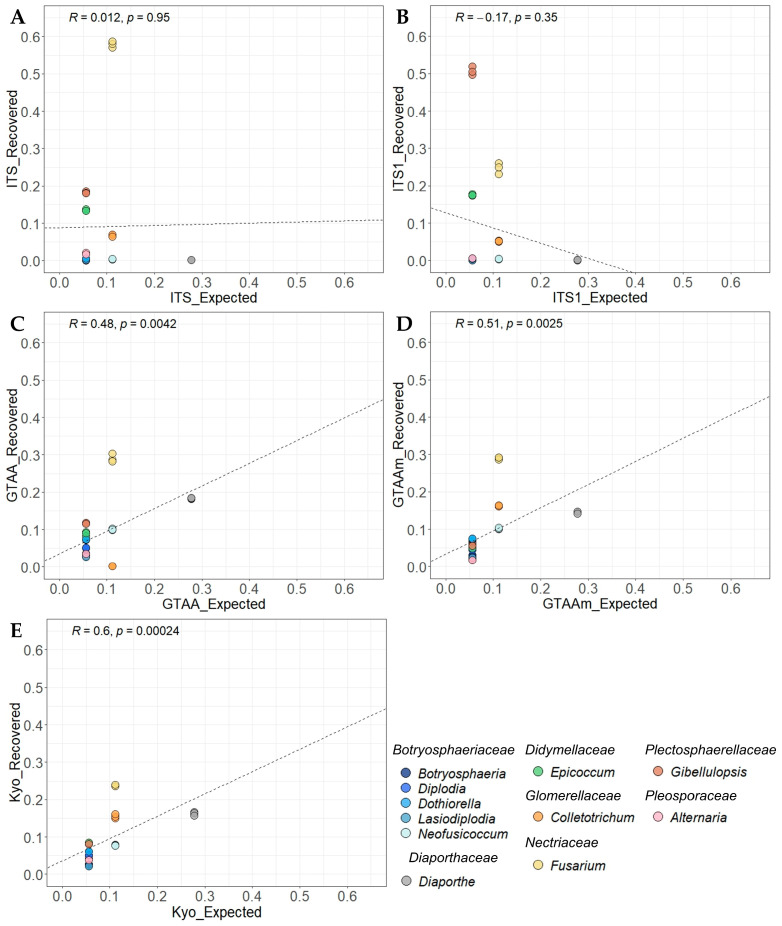
Recovered relative abundances plotted against expected ones at the genus level obtained for Mock4 for each primer set: ITS (**A**), ITS1 (**B**), GTAA (**C**), GTAAm (**D**) and Kyo (**E**). Pearson’s correlation coefficients were calculated based on the relative abundance values of the three replicates.

**Figure 3 plants-12-02383-f003:**
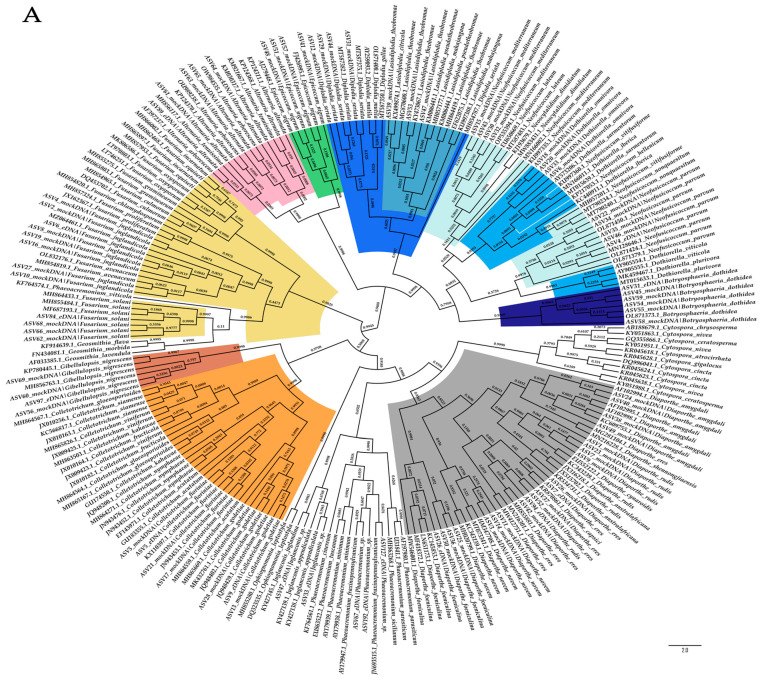
Bayesian inference phylogenetic consensus cladogram gathering ASVs from mock communities (mockDNA) and from environmental samples (eDNA) obtained from metabarcoding sequencing with GTAAm with ITSx extraction (**A**) and Kyo without ITSx extraction (**B**). Bayesian posterior probabilities values are represented at branches. Taxonomic assignment of ASVs from mock communities was based on the local ITS database, and taxonomic assignment of supplementary environmental ASVs (i.e., for *Juglanconis* and *Phaeoacremonium* genera) was based on BLASTn results. Phylogenetic cladograms were rooted with *Alternaria.*

**Figure 4 plants-12-02383-f004:**
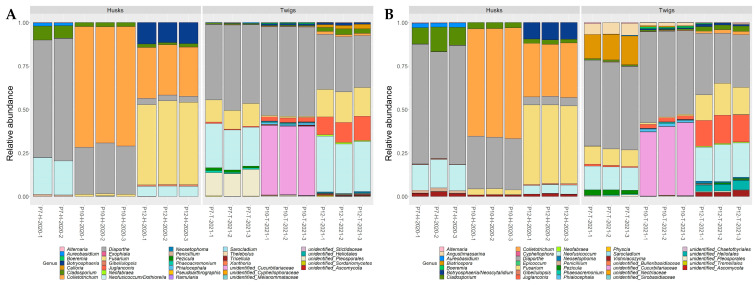
Histograms showing the relative abundance of fungal genera in environmental samples collected in 2020 and 2021 in replicates (−1, −2, −3) of walnut husks (H) and twigs (T) from three orchards (P7, P10, P12). Relative abundances were obtained with metabarcoding sequencing after amplification with GTAAm with ITS2 region extraction (**A**) and Kyo without ITS2 region extraction (**B**).

**Figure 5 plants-12-02383-f005:**
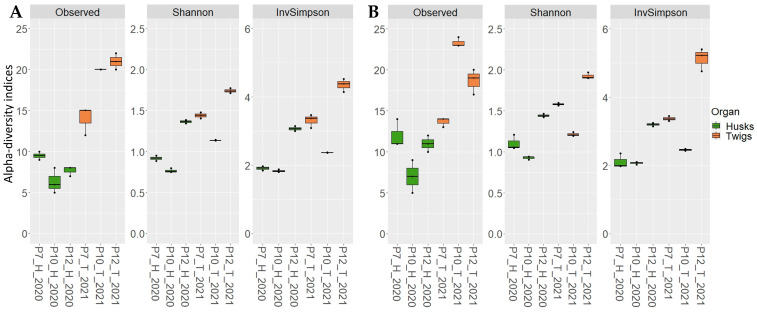
Alpha-diversity indices calculated at the genus level on the rarefied dataset in environmental samples (Husks: H and Twigs: T) from three orchards (P7, P10 and P12) obtained with GTAAm (**A**) and Kyo (**B**) combinations.

**Figure 6 plants-12-02383-f006:**
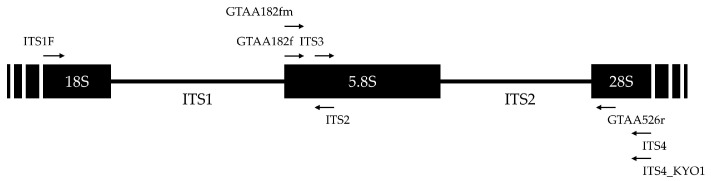
Schematic representation of the fixation sites of forward and reverse primers listed in Table 6 on the ITS region of *N. parvum* (OL639139.1) as an example. Forward primers are represented by right-pointing arrows, and reverse primers by left-pointing arrows.

**Figure 7 plants-12-02383-f007:**
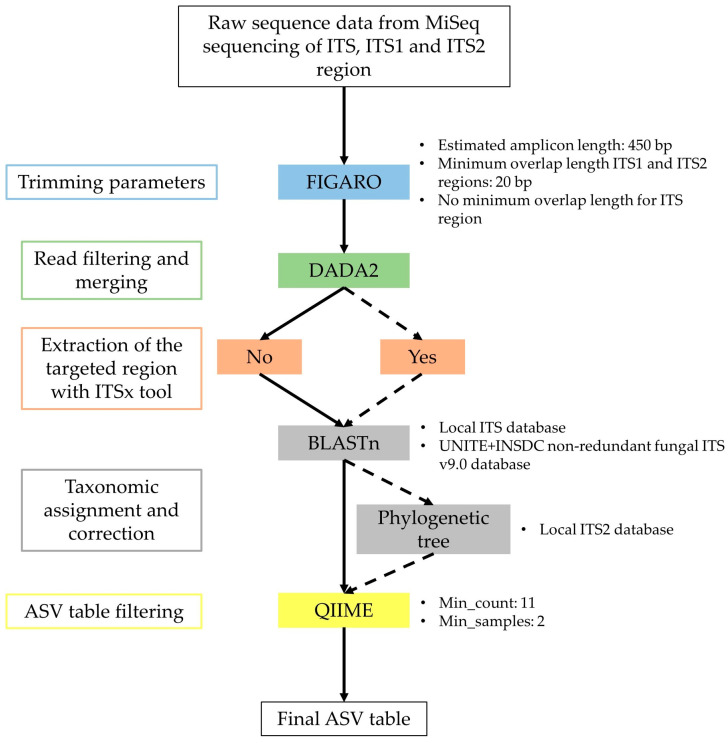
Workflow of bioinformatic steps and tools used in this study during data pre-processing. Parameters are listed at the right of each step. Solid arrows correspond to the basic pipeline initially followed for the five primer pairs used to sequence mock communities. Dashed arrows correspond to optional steps for GTAAm and Kyo for the sequencing of mock communities and environmental samples.

**Table 1 plants-12-02383-t001:** Genus recovery and primer sensitivity on mock communities. Cells were subdivided when results differed between replicates. Green cells correspond to True Positive (TP) ASVs (i.e., detected and well assigned at the genus level), blue and red cells to False Negative (FN) ASVs (i.e., detected but inaccurately assigned at the genus level, or not detected, respectively) and gray cells to species not involved in the mock community.

Fungal Species Included in Mock Communities	Mock1					Mock2					Mock3							Mock4									Mock5						
	ITS	ITS1	GTAA	GTAAm	Kyo	ITS	ITS1	GTAA	GTAAm	Kyo	ITS	ITS1			GTAA	GTAAm	Kyo	ITS			ITS1			GTAA	GTAAm	Kyo	ITS	ITS1			GTAA	GTAAm	Kyo
*Botryosphaeria dothidea*																																	
*Diplodia seriata*																																	
*Dothiorella omnivora*																																	
*Lasiodiplodia theobromae*																																	
*Neofusicoccum mediterraneum*																																	
*N. parvum*																																	
*Diaporthe amygdali*																																	
*Dia. eres*																																	
*Dia. foeniculina*																																	
*Dia. novem*																																	
*Dia. rudis*																																	
*Epicoccum nigrum*																																	
*Colletotrichum fioriniae*																																	
*C. godetiae*																																	
*Fusarium juglandicola*																																	
*F. solani*																																	
*Gibellulopsis nigrescens*																																	
*Alternaria alternata*																																	

**Table 2 plants-12-02383-t002:** Sensitivity, precision and similarity of composition values and number of ASVs for each mock community and each primer pair. Results are presented as mean +/− standard deviation over the three technical replicates when they showed different values. ASVs assigned to *Sarocladium* were not taken into account for the calculation of the performance criteria.

Sample (Number of Expected ASVs)	Primer Set	Sensitivity	Precision	Similarity of Composition	Number of Obtained ASVs
Mock1 (6 ASVs)	ITS	0.833 ^b^	0.045 ^c^ ± 0.001	0.739 ^c^ ± 0.016	113 ± 3.6
	ITS1	0.833 ^b^	0.242 ^abc^ ± 0.007	0.845 ^a^ ± 0.003	21.7 ± 0.6
	GTAA	1 ^a^	0.500 ^a^	0.846 ^ab^ ± 0.001	12
	GTAAm	1 ^a^	0.250 ^ab^	0.820 ^abc^ ± 0.002	24
	Kyo	0.833 ^b^	0.114 ^bc^	0.814 ^bc^ ± 0.004	45
Mock2 (5 ASVs)	ITS	1 ^a^	0.099 ^b^ ± 0.002	1 ^a^	50.3 ± 1.1
	ITS1	1 ^a^	0.341 ^ac^ ± 0.014	1 ^a^	14.7 ± 0.6
	GTAA	1 ^a^	0.500 ^a^	1 ^a^	10
	GTAAm	1 ^a^	0.278 ^abc^	1 ^a^	18
	Kyo	1 ^a^	0.135 ^bc^	1 ^a^	37
Mock3 (17 ASVs)	ITS	0.941 ^a^	0.178 ^ac^ ± 0.003	0.301 ^c^ ± 0.007	90.7 ± 1.5
	ITS1	0.804 ^b^ ± 0.068	0.518 ^b^ ± 0.032	0.324 ^ac^ ± 0.005	27.3 ± 0.6
	GTAA	0.882 ^ab^	0.454 ^ab^	0.710 ^ab^ ± 0.003	34
	GTAAm	0.941 ^a^	0.253 ^abc^ ± 0.002	0.702 ^abc^ ± 0.010	64.3 ± 0.6
	Kyo	0.882 ^ab^	0.115 ^c^	0.749 ^b^ ± 0.005	132
Mock4 (18 ASVs)	ITS	0.870 ^b^ ± 0.032	0.154 ^b^ ± 0.011	0.325 ^ac^ ± 0.003	103.7 ± 4.9
	ITS1	0.685 ^b^ ± 0.085	0.457 ^a^ ± 0.056	0.292 ^c^ ± 0.001	28
	GTAA	0.889 ^ab^	0.453 ^a^ ± 0.015	0.707 ^abc^ ± 0.006	36.3 ± 1.1
	GTAAm	0.944 ^a^	0.260 ^ab^ ± 0.002	0.751 ^ab^ ± 0.006	66.3 ± 0.6
	Kyo	0.889 ^ab^	0.115 ^b^	0.769 ^b^ ± 0.007	140.7 ± 0.6
Mock5 (17 ASVs)	ITS	0.941 ^a^	0.178 ^ac^ ± 0.007	0.312 ^c^ ± 0.006	91 ± 3.5
	ITS1	0.706 ^b^ ± 0.059	0.495 ^b^ ± 0.025	0.327 ^ac^ ± 0.020	25.3 ± 3.0
	GTAA	0.882 ^ab^	0.454 ^ab^	0.731 ^ab^ ± 0.009	34
	GTAAm	0.941 ^a^	0.255 ^abc^ ± 0.002	0.718 ^abc^ ± 0.005	63.7 ± 0.6
	Kyo	0.882 ^ab^	0.117 ^c^ ± 0.004	0.764 ^b^ ± 0.009	129.7 ± 4.0

Lowercase letters indicate significant differences at 0.05 threshold based on Dunn’s test. Statistical comparisons were performed by mock and by column.

**Table 3 plants-12-02383-t003:** Determination of the genus- and species-level resolutions obtained with GTAAm and Kyo with (w/) or without (wo/) ITSx extraction after DADA2 sequence pre-processing and following validation of taxonomic assignment with phylogenetic trees. Green cells correspond to a correct assignment at the genus or species level, blue cells correspond to an inaccurate assignment and gray cells to fungal species absent from our local databases.

Fungal Species Included in Mock Communities	GTAAm				Kyo			
	w/		wo/		w/		wo/	
	Genus	Species	Genus	Species	Genus	Species	Genus	Species
*Botryosphaeria dothidea*								
*Diplodia seriata*								
*Dothiorella omnivora*								
*Lasiodiplodia theobromae*								
*Neofusicoccum mediterraneum*								
*N. parvum*								
*Diaporthe amygdali*								
*Dia. eres*								
*Dia. foeniculina*								
*Dia. novem*								
*Dia. rudis*								
*Epicoccum nigrum*								
*Colletotrichum fioriniae*								
*C. godetiae*								
*Fusarium juglandicola*								
*F. solani*								
*Gibellulopsis nigrescens*								
*Alternaria alternata*								

**Table 4 plants-12-02383-t004:** Sensitivity, precision and similarity of composition values for GTAA182f/GTAA526r (GTAAm) and ITS3/ITS4_KYO1 (Kyo) primer pairs with (w/) or without (w/o) ITS2 extraction at the genus (Gen) and species level (Sp) after manual inspection of taxonomic assignment. Results are presented as mean +/− standard deviation over the three replicates. ASVs assigned to *Sarocladium* were not taken into account for the calculation of the performance criteria.

Sample	Primer Set	Sensitivity				Precision				Similarity of Composition	
		Genus		Species		Genus		Species			
		w/	w/o	w/	w/o	w/	w/o	w/	w/o	Genus	Species
Mock1	GTAAm	0.5 ^a^	0.333 ^b^	0.167 ^a^	0 ^b^	0.143 ^a^	0.1 ^ab^	0.053 ^a^	0 ^b^	0.820 ^A^ ± 0.002	0.822 ^A^ ± 0.004
	Kyo	0.333 ^b^	0.5 ^a^	0.167 ^a^	0.167 ^a^	0.049 ^c^	0.071 ^bc^	0.025 ^ab^	0.025 ^ab^	0.814 ^B^± 0.004	0.826 ^A^ ± 0.006
Mock2	GTAAm	1 ^a^	1 ^a^	0.4 ^a^	0.4 ^a^	0.278 ^a^	0.278 ^a^	0.133 ^a^	0.133 ^a^	1 ^A^	0.790 ^A^ ± 0.004
	Kyo	1 ^a^	1 ^a^	0.5 ^a^	0.6 ^b^	0.135 ^b^	0.135 ^b^	0.059 ^b^	0.086 ^ab^	1 ^A^	0.817 ^B^ ± 0.004
Mock3	GTAAm	0.823 ^a^	0.765 ^b^	0.235 ^ab^	0.176 ^a^	0.228 ^a^ ± 0.002	0.215 ^ab^ ± 0.002	0.078 ^a^ ± 0.001	0.060 ^bc^ ± 0.001	0.702 ^A^ ± 0.010	0.655 ^A^ ± 0.008
	Kyo	0.765 ^b^	0.823 ^a^	0.236 ^ab^	0.470 ^b^	0.101 ^c^	0.108 ^bc^	0.034 ^b^	0.065 ^ac^	0.749 ^B^ ± 0.005	0.745 ^B^ ± 0.001
Mock4	GTAAm	0.833 ^a^	0.779 ^b^	0.278 ^ab^	0.222 ^a^	0.237 ^a^ ± 0.002	0.225 ^ab^ ± 0.002	0.094 ^a^ ± 0.001	0.076 ^ab^ ± 0.001	0.751 ^A^ ± 0.006	0.710 ^A^± 0.003
	Kyo	0.778 ^b^	0.833 ^a^	0.245 ^ab^	0.5 ^b^	0.102 ^c^	0.109 ^bc^	0.039 ^c^	0.068 ^bc^	0.769 ^B^ ± 0.007	0.770 ^B^ ± 0.008
Mock5	GTAAm	0.823 ^a^	0.765 ^b^	0.235 ^ab^	0.176 ^a^	0.229 ^a^	0.219 ^ab^ ± 0.002	0.078 ^a^	0.060 ^bc^ ± 0.001	0.718 ^A^ ± 0.005	0.672 ^A^ ± 0.004
	Kyo	0.765 ^b^	0.823 ^a^	0.235 ^ab^	0.470 ^b^	0.103 ^c^ ± 0.003	0.111 ^bc^ ± 0.003	0.034 ^b^ ± 0.001	0.066 ^ac^ ± 0.002	0.764 ^B^ ± 0.009	0.765 ^B^ ± 0.010

w/: with ITSx extraction; w/o: without ITSx extraction; Genus: genus level; Species: species level. Lowercase letters indicate significant differences at 0.05 threshold based on Dunn’s test. Statistical comparisons were performed by level (genus or species) between primer sets for sensitivity and precision values with and without ITSx extraction for each mock community. Uppercase letters indicate significant differences at 0.05 threshold based on Wilcoxon’s test. Statistical comparisons were performed by level (genus or species) and by mock between primer sets for compositional similarity values.

**Table 5 plants-12-02383-t005:** Fungal species associated with wood and fruit diseases among nut and olive trees.

Classification	Species	Host (Species)				References
		Walnut (*Juglans regia*)	Almond (*Prunus dulcis*)	Pistachio (*Pistacia vera*)	Olive (*Olea europaea*)	
*Bionectriaceae*	*Geosmithia flava*			X		[76]
	*G. lavendula*			X		[76]
	*G. morbida*	X				[77,78]
*Botryosphaeriaceae*	*Botryosphaeria dothidea*	X	X	X	X	[11,12,79,80,81,82,83,84,85]
	*Diplodia gallae*	X				[70]
	*D. mutila*	X	X	X	X	[80,83,84,86,87,88]
	*D. seriata*	X	X	X	X	[7,80,81,82,83,86,87,89,90,91]
	*Dothiorella iberica*	X	X	X	X	[80,83,87,91]
	*Dot. omnivora*	X				[10,70]
	*Dot. plurivora*	X				[70]
	*Dot. sarmentorum*	X	X	X		[7,70,81,91]
	*Dot. viticola*	X	X			[70,87]
	*Lasiodiplodia citricola*	X		X		[70,80,91]
	*L. mahajangana*	X		X		[70,92]
	*L. pseudotheobromae*	X		X		[85,93]
	*L. theobromae*	X	X	X	X	[70,81,83,84,87]
	*Neofusicoccum hellenicum*			X		[92]
	*N. luteum*				X	[83,84]
	*N. mediterraneum*	X	X	X	X	[7,80,81,83,85,87,88,91]
	*N. nonquaesitum*	X	X			[80,81]
	*N. parvum*	X	X	X	X	[7,11,80,81,82,84,85,86,87]
	*N. vitifusiforme*	X	X	X	X	[80,83,87,90,91]
	*Neoscytalidium dimidiatum*	X	X	X	X	[80,87,94,95,96]
*Diaporthaceae*	*Diaporthe amygdali*	X	X			[7,82,97,98]
	*Dia. australafricana*	X	X			[86,87]
	*Dia. biguttulata*	X				[99]
	*Dia. capsici*	X				[100]
	*Dia. cynaroidis*	X				[86]
	*Dia. eres*	X	X			[10,11,87,99]
	*Dia. foeniculina*	X	X	X	X	[7,11,80,85,90,98]
	*Dia. juglandicola*	X				[101]
	*Dia. novem*	X	X			[87], this study
	*Dia. rudis*	X		X	X	[83,91,102], this study
	*Dia. shennongjiaensis*	X				[103]
*Didymellaceae*	*Epicoccum nigrum*	X			X	[11,84]
*Glomerellaceae*	*Colletotrichum acutatum*	X	X	X	X	[104,105,106,107,108]
	*C. fioriniae*	X	X	X	X	[5,11,105,107,109,110,111]
	*C. fructicola*	X				[112]
	*C. gloeosporioides*	X		X	X	[5,107,109,113,114]
	*C. godetiae*	X	X		X	[5,11,106,107,109,110]
	*C. kahawae*	X			X	[109,114]
	*C. nymphaeae*	X	X		X	[5,109,115,116,117]
	*C. siamense*	X		X	X	[108,113,114]
	*C. viniferum*	X				[104]
*Gnomoniaceae*	*Ophiognomonia leptostyla*	X				[118,119]
*Juglanconidaceae*	*Juglanconis appendiculata*	X				[71]
	*J. juglandina*	X				[71,120]
*Nectriaceae*	*Fusarium avenaceum*	X	X	X		[121,122,123,124,125]
	*F. chlamydosporum*			X	X	[125,126]
	*F. culmorum*	X		X		[121,123,125]
	*F. equiseti*			X		[125]
	*F. graminearum*	X				[123]
	*F. incarnatum*	X		X		[123,125,127]
	*F. juglandicola*	X				[128]
	*F. oxysporum*	X		X	X	[121,123,125,126]
	*F. proliferatum*	X		X		[123,129]
	*F. solani*	X	X	X	X	[11,124,125,126,130,131]
*Plectosphaerellaceae*	*Gibellulopsis nigrescens*	X				[11]
*Pleosporaceae*	*Alternaria alternata*	X	X	X	X	[79,84,121,132,133]
	*A. arborescens*		X	X		[132,133]
	*A. tenuissima*	X	X	X	X	[79,84,130,132,133]
*Togniniaceae*	*Phaeoacremonium cinereum*	X		X		[70]
	*P. fraxinopennsylvanicum*	X				[70]
	*P. italicum*	X	X		X	[70,134]
	*P. minimum*	X	X	X	X	[70,85,90]
	*P. parasiticum*	X	X	X	X	[70,90,134]
	*P. sicilianum*	X			X	[10,134]
	*P. tuscanum*	X				[70]
	*P. viticola*	X	X	X	X	[70,135]
*Valsaceae*	*Cytospora atrocirrhata*	X				[136,137]
	*Cyt. californica*	X	X	X		[87,88,136]
	*Cyt. ceratosperma*	X				[137]
	*Cyt. chrysosperma*	X				[136,138]
	*Cyt. cincta*	X				[138]
	*Cyt. gigalocus*	X				[136,137]
	*Cyt. joaquinensis*	X		X		[88,136]
	*Cyt. nivea*	X				[139]
	*Cyt. plurivora*	X	X	X	X	[87,136]

**Table 6 plants-12-02383-t006:** Summary of PCR primers used in this study for metabarcoding sequencing. Amplicon sizes are based on the ITS region of *N. parvum* (OL639139.1) presented in Figure 1.

Barcode	Primer Name	Set Name	Direction	Sequence (5′-3′)	Amplicon Size (bp)	Primer Reference
ITS	ITS1F	ITS	Forward	CTTGGTCATTTAGAGGAAGTAA	617	[37]
	ITS4		Reverse	TCCTCCGCTTATTGATATGC		[36]
ITS1	ITS1F	ITS1	Forward	CTTGGTCATTTAGAGGAAGTAA	295	[37]
	ITS2		Reverse	GCTGCGTTCTTCATCGATGC		[36]
ITS2	GTAA182f	GTAA	Forward	AAAACTTTCAACAACGGATC	337	[20]
	GTAA526r		Reverse	TYCCTACCTGATCCGAGGTC		[20]
	**GTAA182fm**	GTAAm	Forward	AAAACTTTYAACAACGGATC	337	This study
	**GTAA526r**		Reverse	TYCCTACCTGATCCGAGGTC		[20]
	**ITS3**	Kyo	Forward	GCATCGATGAAGAACGCAGC	342	[36]
	**ITS4_KYO1**		Reverse	TCCTCCGCTTWTTGWTWTGC		[38]

In bold, the two primer pairs selected to compare the performance of taxonomic assignment and applied to metabarcoding sequencing of environmental samples.

**Table 7 plants-12-02383-t007:** Species included in mock communities and their origin. Each mock community was composed of DNA solutions of the same concentration in the same proportion.

Classification	Species	Strain No.	DNA Solutions Included in Mocks				
			Mock1 ^e^	Mock2 ^f^	Mock3 ^g^	Mock4 ^h^	Mock5 ^i^
*Plant*							
*Juglandaceae*	*Juglans regia*	-					X
*Fungi*							
*Botryosphaeriaceae*	*Botryosphaeria dothidea*	P12N1I2_2020 ^a^	X		X	X	X
	*Diplodia seriata*	CBS112555 ^b^	X		X	X	X
	*Dothiorella omnivora*	CBS140349 ^b^	X		X	X	X
	*Lasiodiplodia theobromae*	CBS164.96 ^b^	X		X	X	X
	*Neofusicoccum mediterraneum*	CBS121718 ^b^	X		X	X	X
	*N. parvum*	P12N1I1_2020 ^a^	X		X	X	X
*Diaporthaceae*	*Diaporthe amygdali*	CBS126679 ^b^		X	X	X	X
	*Dia. eres*	P12N2I3_2020 ^a^		X	X	X	X
	*Dia. foeniculina*	UBOCC-A-122019 ^c^		X	X	X	X
	*Dia. novem*	UBOCC-A-122020 ^c^		X	X	X	X
	*Dia. rudis*	P9N3I1_2020 ^a^		X	X	X	X
*Didymellaceae*	*Epicoccum nigrum*	P12N5I8_2020 ^a^			X	X	X
*Glomerellaceae*	*Colletotrichum fioriniae*	UBOCC-A-122017 ^c^			X	X	X
	*C. godetiae*	UBOCC-A-122016 ^c^			X	X	X
*Nectriaceae*	*Fusarium juglandicola*	UBOCC-A-119001 ^d^			X	X	X
	*F. solani*	UBOCC-A-122023 ^c^			X	X	X
*Plectosphaerellaceae*	*Gibellulopsis nigrescens*	UBOCC-A-122024 ^c^				X	
*Pleosporaceae*	*Alternaria alternata*	UBOCC-A-122015 ^c^			X	X	X

^a^ Strains isolated from environmental samples and long-time conserved in 20% glycerol at −80 °C at LUBEM laboratory; ^b^ Strains from the CBS collection; ^c^ Strains isolated from environmental samples and added in UBOCC for this study; ^d^ Strain from the UBOCC; ^e^ Six fungal species—equivalent to 16.67% volume each; ^f^ Five fungal species—equivalent to 20% volume each; ^g^ Seventeen fungal species—equivalent to 5.89% volume each; ^h^ Eighteen fungal species—equivalent to 5.56% volume each; ^i^ Seventeen fungal species and one plant species—equivalent to 5.56% volume each.

## Data Availability

The data presented in this study have been deposited at the NCBI and are openly available under the Bioproject PRJNA951625.

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
