# Peer review of "Profiling Walnut Fungal Pathobiome Associated with Walnut Dieback Using Community-Targeted DNA Metabarcoding"

_plants, 2023, doi:10.3390/plants12122383_

Round 1

Reviewer 1 Report

A very valuable study, taking into account the best standards in the field of metagenome analysis.

 However, I have a question whether the title should not be modified. It sounds good, but does it really reflect the scope of work? The pathobiom i.e. a general set of pathogens responsible for diebeck, while the authors focus only on pathogenic fungi.

Author Response

Dear reviewer, we would like to thank you for your comment and question about our manuscript. We fully agree with your concern related to the title which was modified as follows "Profiling walnut fungal pathobiome associated with walnut dieback using community-targeted DNA metabarcoding". In addition, the use of "fungal pathobiome" was added in the text (L. 11 and 282).

Reviewer 2 Report

Dear authors,

I enjoyed to read your paper very much. I wish you success in your future research and will be happy to see new inovation in metabarcoding methods.

Thank you.

Please see the review with recomendations to really minor corrections in the attached file.

Best regards, reviewer.

English language and style are fine/minor spell check required

Author Response

Dear reviewer, we would like to thank you for your careful reading of our manuscript. The corrections have been all taken into account and are directly included in the manuscript and supplementary data using the "Track Changes" mode. All taxonomic ranks were changed from regular to italic style (including in the legends of Figure 1 and Figure 2) and references have been revised and homogenized. Moreover, English language have been revised and edited. 

Line 38

………and more recently in France. Add the citations to the listed countries.

The text has been modified accordingly (L. 38 to 40).

Line 306

…data, underrepresented taxa [66]

Add a full stop after the end of the sentence as [66].

The text has been modified accordingly (L. 333).

Line 595

The paper is already published, fill in the page range.

The text has been modified accordingly (L. 625).

Line 739

Comparison of Metabarcoding Taxonomic Markers to Describe Fungal Communities in Fermented Foods

The full stop is missing behind the title.

The text has been modified accordingly (L. 776).

Line 740

The paper is still not published but the doi number is already available.

See the preprint server: https://www.biorxiv.org/content/10.1101/2023.01.13.523754v1

and add the DOI: 10.1101/2023.01.13.523754

The text has been modified accordingly (L. 777).

Reviewer 3 Report

The study highlights the importance of optimization of metabarcoding parameters and bioinformatics pipeline for the microbiome, including the pathobiome analysis.  It was interesting to realize the percentage of data (species recognition loss) during the pathobiom analysis as well. The manuscript is well structured, tables and figures are informative. Figure 3 with small letters and low resolution was difficult to read.

Author Response

Dear reviewer, we would like to thank you for your careful reading and for your comment about our manuscript. Figure 3 has been split and positioned on two pages instead of one. Also, the resolution of our figures is higher than that included in the manuscript for peer-reviewing.